# Inhibition of miR-25 Ameliorates Cardiac Dysfunction and Fibrosis by Restoring Krüppel-like Factor 4 Expression

**DOI:** 10.3390/ijms241512434

**Published:** 2023-08-04

**Authors:** Cholong Lee, Sunghye Cho, Dongtak Jeong

**Affiliations:** Department of Medicinal & Life Science, College of Science and Convergence Technology, Hanyang University-ERICA, 55 Hanyangdaehak-ro, Sangnok-gu, Ansan 15588, Republic of Korea; choulong@hanyang.ac.kr (C.L.); josunghye@hanyang.ac.kr (S.C.)

**Keywords:** cardiac dysfunction, hypertension, cardiac hypertrophy, cardiac fibrosis, miR-25, KLF4, Angiotensin II, inflammation, SERCA2a

## Abstract

Cardiac hypertrophy is an adaptive response to various pathological insults, including hypertension. However, sustained hypertrophy can cause impaired calcium regulation, cardiac dysfunction, and remodeling, accompanied by cardiac fibrosis. Our previous study identified miR-25 as a regulator of SERCA2a, and found that the inhibition of miR-25 improved cardiac function and reduced fibrosis by restoring SERCA2a expression in a murine heart failure model. However, the precise mechanism underlying the reduction in fibrosis following miR-25 inhibition remains unclear. Therefore, we postulate that miR-25 may have additional targets that contribute to regulating cardiac fibrosis. Using in silico analysis, Krüppel-like factor 4 (KLF4) was identified as an additional target of miR-25. Further experiments confirmed that KLF4 was directly targeted by miR-25 and that its expression was reduced by long-term treatment with Angiotensin II, a major hypertrophic inducer. Subsequently, treatment with an miR-25 inhibitor alleviated the cardiac dysfunction, fibrosis, and inflammation induced by Angiotensin II (Ang II). These findings indicate that inhibiting miR-25 not only enhances calcium cycling and cardiac function via SERCA2a restoration but also reduces fibrosis by restoring KLF4 expression. Therefore, targeting miR-25 may be a promising therapeutic strategy for treating hypertensive heart diseases.

## 1. Introduction

Hypertension is a leading cause of cardiovascular morbidity and mortality world-wide [1]. An early response of the myocardium to hypertension is an increase in the protein level of Ang II, which is associated with cardiac hypertrophy development [2,3]. Cardiac hypertrophy initially serves as an adaptive response, but sustained hypertrophy increases the risk of sudden death and progression to heart failure [4]. Therefore, therapies aimed at inhibiting or reversing cardiac hypertrophy have significant clinical value. Extensive investigations have revealed a complex hypertrophic signaling network with crosstalk between multiple parallel signaling pathways. Impaired calcium uptake is a common feature among the various causes of cardiac hypertrophy. Thus, restoring intracellular Ca^2+^-handling proteins holds promise as a therapeutic strategy for preserving cardiomyocyte performance and preventing the progression of compensatory hypertrophy to heart failure [5,6,7,8]. Our research group has previously focused on SERCA2a, a cardiac-specific calcium-transporting ATPase, because its reduced expression and activity are hallmarks of heart failure [5].

MicroRNAs (miRNAs) are a family of small non-coding RNAs that regulate gene ex-pression at the post-transcriptional level by silencing mRNA [9]. In a previous study, our group identified miR-25 as a significant suppressor of SERCA2a, with pathological up-regulation of endogenous miR-25 observed in myocardial samples obtained from patients with severe heart failure.

In the same study, we found that miR-25 inhibition halts the progression of cardiac dysfunction in mice undergoing transaortic constriction (TAC) surgery [7]. This effect was primarily attributed to the restoration of SERCA2a mRNA and protein levels. Additionally, miR-25 inhibition significantly reduced cardiac fibrosis in the same animal model; however, the underlying mechanism remained unclear. This led us to hypothesize that miR-25 may have additional targets for regulating fibrosis.

To identify the potential targets of miR-25 in cardiac fibrosis, we performed an in silico analysis using three miRNA target identification software packages: PicTar (PicTar predictions in vertebrates (Krek et al., 2005) and flies (Grün et al., 2005)), TargetScan7 (TargetScanHuman 7.0), and miRDB (http://www.mirdb.org/ accessed on 18 May 2022). This analysis identified 145 new targets, including Krüppel-like factor 4 (KLF4).

KLF4 is a zinc finger-containing transcription factor involved in various cellular processes, including those of the cardiovascular system [10]. It regulates cell fate and differentiation [11], and can either promote or suppress the growth of different types of tumor cell [12]. However, the role of KLF4 in tissue fibrosis is still limited and controversial [13,14]. For example, its expression was reported to be diminished in fibrotic lungs, where it suppressed the TGF-β-induced epithelial–mesenchymal transition (EMT) to myofibroblasts in a model of bleomycin-induced pulmonary fibrosis [13]. In contrast, it activated Hippo signaling, and promoted renal EMT and renal fibrosis in an ischemia–reperfusion model [14]. Furthermore, KLF4 has been shown to both inhibit [15] and enhance [16] fibroblast differentiation into myofibroblasts in response to TGF-β. KLF4 also promotes M2 polarization and TGF-β secretion by macrophages [17,18]. Thus, KLF4 has the potential to exert contrasting effects pertinent to fibrogenesis in distinct cell types, but its role in cardiac fibroblasts remains unknown.

To elucidate the role of KLF4 in cardiac hypertrophy and fibrosis, we used two well-established models. In the in vitro study, we used H9c2 myoblasts to investigate the effect of miR-25 inhibition under Ang II treatment. For in vivo characterization, we administered a subcutaneous Ang II infusion to a mouse model using an osmotic minipump, along with AAV9 miR-25 TuD inhibitor delivery.

Collectively, in this study, we examined the effect of miR-25 inhibition on Ang II-induced cardiac dysfunction and fibrosis both in vitro and in vivo by focusing on the regulation of KLF4 expression.

## 2. Results

### 2.1. KLF4 Is a Newly Identified Target of miR-25 in the Heart

In a previous study, we identified miR-25 as a key regulator of SERCA2a, an important protein involved in calcium handling in cardiomyocytes. We demonstrated that miR-25 inhibition improves cardiac function and reduces fibrosis by restoring SERCA2a expression in a heart failure model. However, the mechanism by which miR-25 inhibition reduces fibrosis was not completely understood. Therefore, we hypothesized that miR-25 may have additional targets that specifically regulate cardiac fibrosis.

To test this hypothesis, we conducted in silico screening using three miRNA target identification software packages: PicTar (PicTar predictions in vertebrates (Krek et al., 2005) and flies (Grün et al., 2005)), TargetScan7 (TargetScanHuman 7.0), and miRDB (http://www.mirdb.org/ accessed on 18 May 2022). A total of 145 potential candidates were identified (Figure 1A). Among these, KLF4 was of particular interest because it is highly conserved across species (Figure 1B) and multiple studies have shown that KLF4 inhibits cardiac hypertrophy and fibrosis by regulating TGF-β [19,20,21,22,23,24], although a contradictory result was also reported [16]. 

To confirm the direct interaction between miR-25 and KLF4, we performed a luciferase reporter assay using a construct containing the KLF4 3’-UTR sequence. We observed significant repression of luciferase activity in the presence of pre-mir-25 in a dose-dependent manner, indicating that miR-25 directly targets the 3’-UTR region of KLF4 (Figure 1C). To validate this finding, we conducted co-transfection experiments using pre-miR-25 and miR-25 TuD inhibitors. Notably, the inhibitory effect of pre-miR-25 on KLF4 expression was completely abolished when co-transfected with miR-25 TuD, confirming the specificity of miR-25-mediated KLF4 regulation (Figure 1D). These results provide strong evidence for the direct targeting of KLF4 by miR-25.

To investigate the functional roles of miR-25 and KLF4 in cardiac hypertrophy and fibrosis, H9c2 myoblasts were treated with Ang II. As expected, Ang II treatment significantly increased the expression of hypertrophy and fibrosis markers, including β-MHC, Periostin, and BNP. Simultaneously, the expression of KLF4, a negative regulator of cardiac hypertrophy and fibrosis, was notably decreased (Figure 1E).

However, when H9c2 myoblasts were pretreated with the miR-25 TuD inhibitor, the Ang II-induced upregulation of hypertrophy and fibrosis markers was prevented, and the expression of KLF4 was restored to normal levels (Figure 1E). This suggests that miR-25 inhibition counteracts the effects of Ang II on cardiac hypertrophy and fibrosis by restoring KLF4 expression (Figure 1E).

We also examined the expression of different forms of miR-25 (primary, precursor, and mature) in the same experimental groups. Ang II treatment significantly increased the expression of all three forms of miR-25. However, the delivery of the miR-25 TuD inhibitor effectively repressed the expression of these miR-25 forms induced by Ang II (Figure 1F).

Taken together, these findings provide additional evidence for the regulatory role of miR-25 in cardiac hypertrophy and fibrosis by regulating KLF4 expression in vitro.

### 2.2. miR-25 TuD Transfer Prevents Ang II-Induced Cardiac Dysfunction

To evaluate the impact of miR-25 TuD on cardiac hypertrophy in vivo, we subcutaneously implanted an osmotic minipump for Ang II infusion (3 mg/kg/day) in 8-week-old mice, as illustrated in Figure 2A. To ensure the sustained and long-term inhibition of miR-25, AAV9 miR-25 TuD was systemically delivered via tail vein injection. Four weeks after injection, we assessed the cardiac function using echocardiography; representative images are shown in Figure 2B.

Mice treated with AAV9 EGFP exhibited a significant reduction in fractional shortening (FS) and ejection fraction (EF), indicating impaired cardiac function (Figure 2C and Appendix A). In contrast, mice that received AAV9 miR-25 TuD showed pre-served cardiac function, with FS and EF values comparable to those in the control group. These findings suggest that the inhibition of miR-25 using miR-25 TuD can protect against Ang II-induced cardiac dysfunction in vivo.

### 2.3. Inhibition of miR-25 Normalizes Calcium Signaling and Cardiac Stress Markers

After assessing cardiac function, the mice were sacrificed, and their hearts were removed to examine the molecular markers associated with calcium signaling and cardiac stress. To validate the observed improvement in cardiac function, we measured the levels of calcium-handling proteins, including SERCA2a, NCX1, phosphorylated CaMKII (p-CaMKII), and phosphorylated PLN (p-PLN), and cardiac stress markers, such as ANP, BNP, and β-MHC. Western blotting and qRT-PCR were used to evaluate the protein and mRNA expression levels, respectively.

The results showed that the protein expression levels of SERCA2a and p-PLN were significantly restored in the hearts treated with miR-25 TuD, indicating improved calcium handling (Figure 3A). Conversely, the levels of NCX1 and p-CaMKII were substantially decreased, suggesting reduced calcium overload and attenuated cardiac stress in response to miR-25 inhibition. Furthermore, the protein and mRNA expression levels of ANF, BNP, and β-MHC, which are markers of cardiac stress, were markedly downregulated in miR-25 TuD-treated hearts (Figure 3B,C). Long-term delivery of the miR-25 TuD inhibitor using AAV9 efficiently suppressed the expression of all three miR-25 forms induced by Ang II infusion (Figure 3D). These findings indicate that miR-25 inhibition alleviates cardiac stress and improves calcium handling in the hearts of mice.

### 2.4. Restoration of KLF4 Expression via miR-25 TuD Delivery Ameliorates Cardiac Fibrosis

Given the known role of KLF4 in regulating cardiac fibrosis [19,20,21,22,23,24], we examined the expression of fibrosis markers, including TGF-β, phosphorylated Smad2 (p-Smad2), Periostin, α-SMA, Vimentin, and Fibronectin. In our study, we observed a significant reduction in the expression of all pro-fibrotic markers upon AAV9 miR-25 TuD transfer (Figure 4A). Notably, the expression levels of KLF4 as well as Smad7, an endogenous inhibitor of TGF-β-Smad signaling and potential target of miR-25, were also substantially restored, further supporting the role of miR-25 inhibition in mitigating fibrosis.

To corroborate these findings, we performed histological assessments using Masson’s trichrome (MT), hematoxylin–eosin (H&E), and wheat germ agglutinin (WGA) staining. The results revealed a significant reduction in the fibrotic area, myocyte cross-sectional area, and heart weight-to-body weight (HW/BW) ratio after AAV9 miR-25 TuD delivery (Figure 4B–E). These findings confirm a marked reduction in cardiac fibrosis and hypertrophy.

Interestingly, we also observed that mononuclear cell infiltration was remarkably reduced upon AAV9 miR-25 TuD transfer in the H&E staining results, suggesting that the inflammatory response was also regulated. As KLF4 is known to regulate inflammation [25,26,27], we evaluated the expression of representative inflammatory molecules to assess the impact of miR-25 inhibition on inflammation.

### 2.5. miR-25 TuD Transfer Prevents Cardiac Inflammation via KLF4 Expression

Fibrosis and inflammation are closely linked processes that occur concurrently in various tissues, including the heart [28,29,30]. Inflammation is the initial response to tissue injury or infection. However, persistent or chronic inflammation can contribute to fibrosis development. During the inflammatory response, immune cells release pro-fibrotic factors, such as TGF-β, IL-6, and TNF-α, which stimulate fibroblasts to differentiate into myofibroblasts. Myofibroblasts are responsible for the excessive production of ECM components, such as collagen, leading to fibrosis.

In our study, considering the anti-inflammatory effects of KLF4 in different tissues [31,32,33], and that mononuclear cell infiltration was reduced upon miR-25 TuD delivery in our histological data, we hypothesized that miR-25 inhibition may also alleviate cardiac inflammation in the context of Ang II-induced cardiac hypertrophy. To test this hypothesis, we examined the expression of pro-inflammatory proteins, including Galectin-3 and MCP-1, in the same mouse heart tissue. Our results showed that both proteins were significantly upregulated by Ang II infusion; however, their expression was markedly attenuated by AAV9 miR-25 TuD transfer (Figure 5A). Additionally, we quantified the expression of representative inflammatory cytokines such as IL-1β, IL-18, IL-4, and RANTES using qRT-PCR, and found that all cytokines were significantly inhibited by miR-25 TuD delivery (Figure 5B).

Taken together, these findings suggest that miR-25 inhibition mitigates cardiac inflammation by restoring KLF4 expression. This implies that miR-25 inhibition targets fibrosis, as well as exerting anti-inflammatory effects, further highlighting its potential therapeutic value in cardiac hypertrophy and related disorders.

## 3. Discussion

Hypertensive heart disease (HHD) is a significant contributor to cardiovascular morbidity and mortality worldwide [1,34]. Patients on HHD experience sustained pressure overload, leading to decompensated cardiac hypertrophy and heart failure [34]. However, the mechanisms underlying HHD remain unclear.

Calcium dysregulation is considered one of the major consequences of HHD. Previously, several signaling molecules and miRNAs were characterized in AngII-mediated cardiac hypertrophy and remodeling in the context of calcium homeostasis. For instance, miR-1, 133a, and miR-150 are known to be negatively regulate cardiac hypertrophy through the modulation of the calcineurin-NFAT signaling pathway and p300 [35,36,37]. In contrast, miR-155, miR-195, and miR-214 are reported as pro-hypertrophic miRNAs that positively regulate SOCS1, MFN2, and SIRT3, respectively [38,39,40]. Furthermore, both store-operated and receptor-operated calcium entries, via multiple TRP channels, are also involved in Ang II-induced cardiomyocyte hypertrophy [41].

Cardiac fibrosis is a condition characterized by an abnormal accumulation of fibrous connective tissue in the heart. Initially, it results from a repair process in response to injury or stress, such as chronic inflammation or damage due to conditions like hypertension, heart attacks, or cardiomyopathy. However, the excessive and uncontrolled deposition of fibrous tissue in the heart can lead to stiffening and thickening of the cardiac muscle, disrupting its normal function [42]. Multiple miRNAs and signaling molecules have also been identified to regulate cardiac fibrosis. For example, miR-21 has been validated as an activator of cardiac fibroblasts post-MI, subsequently eliciting cardiac fibrosis [43,44]. In addition, Wang et al. reported that miR-92a [45] and miR-195 [46] act as transcriptional regulators of SMAD7, an inhibitor of α-SMA, which is a well-established marker of myofibroblast activation. miR-34a also modulates cardiac fibrosis after MI by targeting SMAD4 [47]. Furthermore, it is also known that the canonical and non-canonical TGF-β pathways have been intensively studied to understand the underlying molecular mechanism of cardiac remodeling and fibrosis [48]. However, the role of miR-25 has not been clearly elucidated in terms of cardiac hypertrophy and fibrosis. For this reason, we sharply focused on the role of miR-25 in the present study. Previously, we identified miR-25 as a specific miRNA targeting SERCA2a, which was significantly upregulated in patients with heart failure, as well as in experimental heart failure models. We also demonstrated that inhibiting miR-25 using antagomirs and tough decoys had beneficial effects on cardiac function and fibrosis [5,7,8]. However, the precise mechanism through which miR-25 inhibition reduces fibrosis remains unclear. Therefore, we hypothesized that miR-25 might have additional targets that specifically regulate fibrosis.

To investigate this, we conducted in silico screening using three miRNA target identification software packages: PicTar (PicTar predictions in vertebrates (Krek et al., 2005) and flies (Grün et al., 2005)), TargetScan7 (TargetScanHuman 7.0), and miRDB (http://www.mirdb.org/ accessed on 18 May 2022). We thus identified 145 potential candidates, among which Krüppel-like factor 4 (KLF4) emerged as a highly conserved target across different species (Figure 1A,B).

KLF4 is a zinc-finger transcription factor that plays a crucial role in various tissues, including the cardiovascular system [49]. For example, KLF4 acts as an antitumorigenic factor, one of the four pluripotency genes used for iPSC reprogramming, and a modulator of macrophage polarization, etc. [12,18,50,51]. Interestingly, previous findings regarding its involvement in cardiac hypertrophy and fibrosis have yielded somewhat contradictory results [13,14,52].

For instance, Liao et al. demonstrated that KLF4 is induced by hypertrophic stimuli in cultured cardiomyocytes and the mouse heart. Moreover, KLF4 overexpression inhibits hypertrophy both in vitro and in vivo [52]. In contrast, Zhang et al. reported that KLF4 enhances the expression of α-SMA and collagen, thus promoting myofibroblast formation through the transcriptional upregulation of TGF-β1 [16].

Based on these inconclusive findings, we first verified whether KLF4 was a direct target of miR-25 using a luciferase reporter system. As depicted in Figure 1C, the luciferase activity of KLF4 was significantly repressed in a dose-dependent manner upon pre-mir-25 treatment. To further validate this result, we performed co-transfection experiments using pre-miRs and miR-25 TuD inhibitors. As expected, the inhibitory effect of pre-miR-25 on KLF4 expression was abolished by co-transfection with miR-25 TuD (Figure 1D).

Next, we investigated the effect of the miR-25 TuD inhibitor on KLF4 expression in Ang II-treated H9c2 myoblasts. Treatment with Ang II resulted in significantly increased expression of representative markers of cardiac remodeling and fibrosis markers, such as β-MHC, Periostin, and BNP, whereas KLF4 expression was substantially decreased. However, pre-treatment with miR-25 TuD prevented the upregulation of these marker genes and restored KLF4 expression, as demonstrated in Figure 1E.

Based on these encouraging results, we performed in vivo experiments using subcutaneous Ang II infusion via osmotic minipumps, as depicted in Figure 2A. Cardiac function was assessed using echocardiography; cardiac dysfunction was found to occur four weeks after Ang II infusion. However, the delivery of AAV9 miR-25 TuD significantly preserved the ejection fraction (%) and fractional shortening (%), as shown in Figure 2B,C.

Subsequently, we evaluated several molecular signaling pathways as well as histological remodeling markers. Consistent with the cardiac functional data, the phosphorylation of calcium signaling molecules such as SERCA2a, PLN, and CaMKII was significantly normalized and NCX expression was reduced upon AAV9 miR-25 TuD transfer, as depicted in Figure 3A. Furthermore, cardiac stress markers, including ANF, BNP, and β-MHC, were measured at the protein and mRNA levels, and their expression was markedly reduced, as shown in Figure 3B,C. Finally, we assessed multiple pro-fibrotic markers such as TGF-β, p-Smad2, Periostin, α-SMA, Vimentin, and Fibronectin, and found that these were substantially inhibited by AAV9 miR-25 TuD treatment (Figure 4A).

In contrast, the expression of KLF4 and Smad7, an endogenous inhibitor of the TGF-β-Smad signaling pathway, were significantly restored by AAV9 miR-25 TuD transfer, indicating inhibition of the pro-fibrotic signaling pathway (Figure 4A). Furthermore, histological analysis supported these observations, as the fibrotic area, myocyte cross-sectional area, and heart weight-to-body weight (HW/BW) ratio were significantly reduced upon AAV9 miR-25 TuD delivery (Figure 4B–E). Considering that fibrosis is closely associated with inflammatory signaling pathways [28,29,30], we evaluated several inflammatory markers, including Galectin-3, MCP-1, IL-1β, IL-18, IL-6, and RANTES. Remarkably, the expression of these markers was markedly reduced in the AAV9 miR-25 TuD group (Figure 5A,B).

Lastly, there are several limitations to performing translational research via the current approach used in this manuscript. First, although AAV has been reported to show minimal immune responses, virtually neutralizing antibodies (NAbs) that recognize all AAV serotypes can be found in a large proportion of the human population (ranging from 30 to 60%). Therefore, AAVs have been genetically engineered to avoid this issue by increasing AAV transduction efficiency or vector tropism [53]. Second, there is a possible off-target expression of anti-oligonucleotide delivery, even with cardiotropic AAV serotype 9. In order to overcome this issue, we may use a tissue-specific promoter or a newly reconstructed cardiotropic vector. Nevertheless, AAV-mediated gene delivery was recently approved for the treatment of several inherited diseases, and it is the most promising delivery method in pre-clinical and clinical settings.

In conclusion, our data demonstrate that the inhibition of miR-25 using a TuD decoy inhibitor ameliorates cardiac dysfunction and fibrosis by restoring SERCA2a and KLF4, both in vitro and in vivo (Figure 5C). These findings suggest that targeting miR-25 and its downstream signaling pathways, including KLF4, may represent a promising strategy to develop novel therapies for hypertensive heart disease.

## 4. Materials and Methods

### 4.1. Cell Culture and Treatment

The H9c2 myoblast cell line, derived from embryonic rat heart, was obtained from ATCC (Manassas, VA, USA). Cells were grown in DMEM (GenDepot, Katy, TX, USA) with 10% fetal bovine serum (FBS), 1% penicillin (100 U/mL), and 1% streptomycin (100 μg/mL), at 37 °C in 5% CO_2_ and 95% air, at a relative humidity of 95%; they were split 1:4 at sub-confluence (80%). Before each experiment, the cells were seeded in six-well plates at a 5 × 10^4^ cells/cm^2^ density and starved for 18 h in serum-free DMEM. H9c2 cells were treated with Ang II (Bachem, Torrance, CA, USA), with or without miR-25 TuD plasmid transfection, for 2 days. These were then used to verify the expression of miR-25, β-MHC, BNP, and KLF4. HEK-293T cells were grown in DMEM supplemented with 10% FBS, 1% penicillin (100 U/mL), and 1% streptomycin (100 μg/mL).

### 4.2. KLF4 3’ UTR Luciferase Assay

The pMIR-Report-Luc-KLF4-FL vector (Plasmid #34597) was purchased from Addgene (Watertown, MA, USA). HEK-293T cells were used for the luciferase assay. Cells were transfected with the plasmids described in Figure 1B using a Lipofectamine 3000 (Invitrogen, Waltham, MA, USA) transfection reagent. After 24 h, the cells were harvested using the lysis buffer from the luciferase assay kit (E1500, Promega, Madison, WI, USA). Luciferase activity was measured using a luminescence microplate reader (Synergy H1; BioTek, Shoreline, WA, USA).

### 4.3. Animals and Experimental Protocol

All experimental procedures were reviewed and approved by the Animal Care Committee of Hanyang University—ERICA (protocol no. HY-IACUC-23-0029). Eight-week-old male C57BL/6 mice, weighing approximately 20–25 g, were purchased from Orient Bio (Seongnam, Republic of Korea). Animals were housed five per cage under specific pathogen-free (SPF) conditions maintained at 22  ±  0.5 °C with an alternating 12 h light–dark cycle. Food and water were provided ad libitum. Animals were allowed to acclimatize to the laboratory for one week before beginning the experiments. To reduce variation, all experiments were performed during the light phase of the cycle. The mice were randomly assigned to the control or experimental groups in cage units. For the prevention model, either AAV9 EGFP or AAV9 miR-25 TuD was first intravenously injected into the mice, and then, AngII (3 mg/kg/day) was administered via subcutaneous implantation of an osmotic minipump (Alzet 1002, Durect Corporation, Cupertino, CA, USA). Four weeks later, cardiac function was evaluated using echocardiography, and all mice were sacrificed for molecular and histological experiments.

### 4.4. Histological Studies

The cardiac tissues were fixed in 4% formaldehyde, passaged, and embedded in paraffin. The paraffin blocks were then sectioned to 6 μm thickness for hematoxylin–eosin (H&E), Masson’s trichrome (MT), and wheat germ agglutinin (WGA) staining. To digitize the selected H&E- and MT-stained slides, a 3DHistech Hi-Scope slide scanner system was used. The WGA-stained slides for the CSA (cross-sectional area) were imaged using a Leica DFC295 color camera (Leica, Berlin, Germany) and analyzed using ImageJ software (Version 1.54d, NIH, Bethesda, MD, USA) at 20× magnification. At least 30 cardiomyocytes per heart were measured. The fibrotic area was calculated as the ratio of the total area of fibrosis to the total area of the section using ImageJ software (Version 1.54d, NIH, Bethesda, MD, USA).

### 4.5. RNA Extraction and cDNA Synthesis

Total RNA was extracted from mouse heart tissue and H9c2 cells using the Hybrid-R miRNA Isolation Kit (GeneAll, Seoul, Republic of Korea) based on the protocol provided by the manufacturer. Briefly, 20 mg heart tissue or 1 × 10^6^ cells were homogenized in 500 μL RiboEx^TM^ TRIzol reagent and 100 μL chloroform was added to each sample. All samples were centrifuged at 12,000× *g* for 15 min at 4 °C to separate the mixture. RNA in the aqueous phase was precipitated with 1 volume (usually 250 μL) of 50% ethanol. All the mixtures were transferred to a mini column (Column Type B) and centrifuged at 10,000× *g* for 30 s at room temperature. After this step, large RNA was bound to the mini column and small (micro) RNA was able to pass through. Small (micro) RNAs were purified via the application of additional Column Type W, and Large RNAs were eluted directly from Column Type B after the washing procedures. RNA purity was determined in a 260/280 nm ratio using a Multiskan SkyHigh NanoDrop spectrophotometer (Thermo Fisher Scientific, Rockford, IL, USA), and only those samples with a ratio of between 1.8 and 2.1 were used in the present study. Reverse transcription (RT) for miRNAs was performed with 4 µg of total RNA in a total reaction volume of 20 µL using the Mir-X miRNA First-Strand Synthesis Kit (Takara Bio Inc., Shiga, Japan), and cDNAs from total RNA was analyzed using amfiRivert cDNA Synthesis Platinum Master Mix (GenDepot, Katy, TX, USA), according to the manufacturers’ protocols. Following cDNA synthesis, all cDNA samples were diluted 10 times in molecular-grade water and stored at −20 °C.

### 4.6. Quantitative Real-Time Polymerase Chain Reaction (qRT-PCR)

All quantitative real-time PCR amplifications were performed using Quantstudio 1 (Applied Biosystems, Waltham, CA, USA). The relative amounts of primary miR-25, precursor miR-25, and mature miR-25-3p were normalized to U6 small nuclear RNA. The mRNA levels of β-MHC, ANF, BNP, IL-18, IL-1β, IL-6, and RANTES were also evaluated, and the results were normalized to 18S ribosomal RNA. The qPCR reaction contained 10 µL 2X SYBR Green Master Mix (DQ384-40h; BioFact, Seoul, Republic of Korea), the forward and reverse primers, RNase-free water, and 2 µL cDNA template per reaction in a final volume of 20 µL. The thermocycling conditions for quantitative PCR were 1 cycle at 50 °C for 2 min and 95 °C for 10 min, and 40 cycles of 30 s at 94 °C and 30 s at 60 °C. All experiments were carried out at least three times. The data were analyzed using QuantStudio Design and Analysis Software v.1.5.2. and the ΔΔC_T_ method was used to calculate the relative expression of the sample gene. The relative quantification (RQ) of gene expression was analyzed using the 2^−ΔΔC^_T_ method. RQ = 2^−ΔΔC^_T_ (C_T_ indicates the cycles required by the fluorescence signal intensity to reach the threshold value in the PCR amplification process; ΔC_T_ sample = C_T_ sample − C_T_ U6 or 18S sample, ΔC_T_ control = C_T_ control − C_T_ U6 or 18S control, ΔΔC_T_ = ΔC_T_ sample − ΔC_T_ control). The primers used for qRT-PCR are listed in Appendix A.

### 4.7. Western Blot Analysis

Membrane and tissue homogenates were prepared as previously described [5]. Proteins were resolved using SDS-PAGE and transferred to nitrocellulose membranes (GenDepot, Katy, TX, USA). Protein bands were detected using standard laboratory protocols. The membranes were incubated with antibodies against KLF4 (1:5000, 11880-1-AP, Proteintech, Chicago, IL, USA), β-MHC (1:5000, ab172967, Abcam, Trumpington, Cambridge, UK), BNP (1:3000, ab239510, Abcam, Trumpington, Cambridge, UK), ANF (1:3000, ab262703, Abcam, Trumpington, Cambridge, UK), Periostin (1:5000, NBP1-30042, Novus, Centennial, CO, USA), SERCA2a (1:20,000, A010-23L, Badrilla, Leeds, UK), NCX1 (1:5000, MA3-926, Invitrogen, Carlsbad, CA, USA), pPLB (S16) (1:5000, A010-12AP, Badrilla, Leeds, UK), total PLB (1:5000, A010-14, Badrilla, Leeds, UK), p-CaMKⅡ (T286) (1:5000, P-247, SIGMA, Saint Louis, MO, USA), Fibronectin (1:5000, 15613-1-AP, Proteintech, Chicago, IL, USA), TGF-β (1:5000, #3711, Cell Signaling, Danvers, MA, USA), p-SMAD2 (1:5000, MA5-15122, Invitrogen, Carlsbad, CA, USA), SMAD2/3 (1:5000, #5678S, Cell Signaling, Danvers, MA, USA), SMAD7 (1:5000, LS-C393154, LS-Bio, Seattle, WA, USA), α-SMA (1:5000, A5228, Sigma, Saint Louis, MO, USA), Vimentin (1:5000, #5741, Cell Signaling, Danvers, MA, USA), Galectin-3 (1:5000, 14-5301-82, Invitrogen, Carlsbad, CA, USA), MCP-1 (1:3000, NBP1-07035, Novus, Centennial, CO, USA), GAPDH (1:5000, #2118, Cell Signaling, Danvers, MA, USA), and α-tubulin (1:5000, ab4074, Abcam, Trumpington, Cambridge, UK). The protein bands were visualized using an ATTO LuminoGraph III Lite Chemiluminescence Imaging System (Atto, Tokyo, Japan). Western blots were analyzed and quantified using CSAnalyzer4 software (Version 2.4.5).

### 4.8. Echocardiographic Assessment

The mice were anesthetized with 2% isoflurane in pure medical oxygen. Two-dimensional images and M-mode tracings were recorded on the short axis at the left ventricular papillary muscle level using a 14.0 MHz transducer to determine the percentage of fractional shortening and ventricular dimensions (Vivid S60N; GE HealthCare, Chicago, IL, USA). Measurements were performed at least three times for each mouse and the average of the measurements was used.

### 4.9. Statistical Analysis

All experiments were carried out in triplicate, and the data are presented as the mean ± standard error of the mean (SEM) for each experiment. GraphPad Prism version 9 (GraphPad Software, Inc., La Jolla, CA, USA) was used for statistical analyses. One-way ANOVA and Student’s *t*-test were used to compare the groups, and * *p* < 0.05, ** *p* < 0.01, *** *p* < 0.001, and **** *p* < 0.0001 were considered statistically significant at different levels.

## Figures and Tables

**Figure 1 ijms-24-12434-f001:**
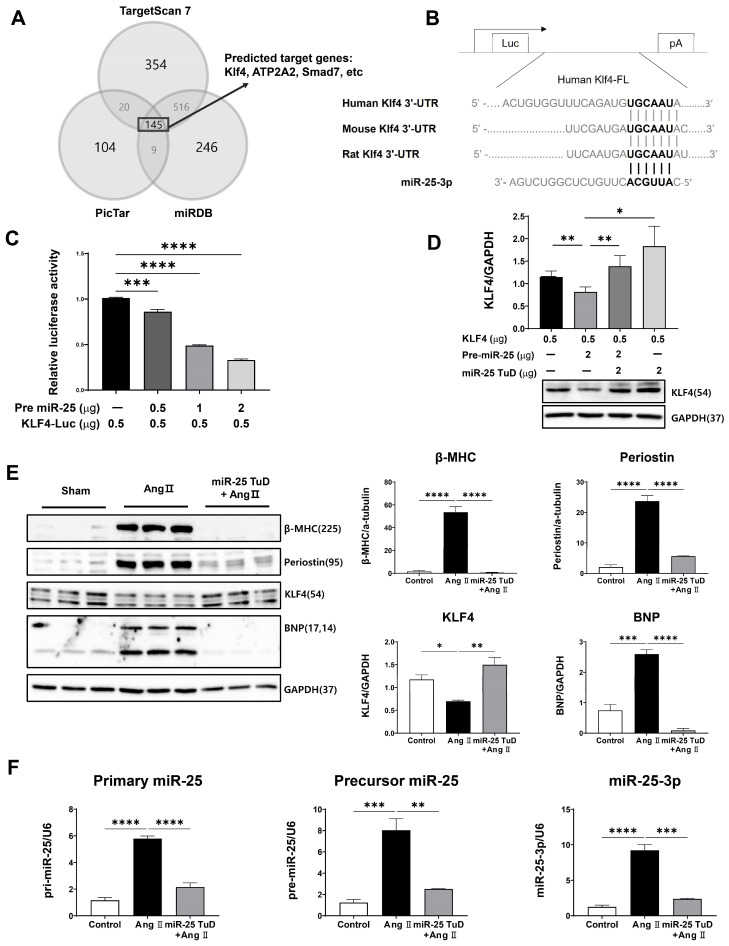
KLF4 is a newly identified target of miR-25 in the heart. (**A**) In silico prediction of the target genes of miR-25-3p. (**B**) The putative miR-25-3p binding site in the 3′ UTR region of KLF4 mRNA was predicted using TargetScan. In addition, the sequence alignment results of the miR-25 binding site on KLF4 mRNA among different species are shown. (**C**) The pre-miR-25 and KLF4-Luc plasmids were transfected into HEK-293T cells. KLF4-Luciferase activity was measured after dose-dependent transfection of pre-miR-25. (**D**) Protein expression of KLF4 was analyzed in HEK-293T cells transfected with KLF4-Luc, pre-miR-25, or miR-25 TuD plasmids. (**E**) H9c2 myoblast cells were transfected with or without the miR-25 TuD plasmid, and Ang II (1 μM) was added for 48 h. Protein expression levels of β-MHC, Periostin, KLF4, and BNP were analyzed using Western blotting. (**F**) Primary miR-25, precursor miR-25, and mature miR-25 levels were analyzed using qPCR (*n* = 3 for each group). Data are presented as mean ± standard error of the mean (SEM). Western blots were analyzed and quantified using CSAnalyzer4 software (Version 2.4.5) (* *p* < 0.05, ** *p* < 0.01, *** *p* < 0.001, **** *p* < 0.0001).

**Figure 2 ijms-24-12434-f002:**
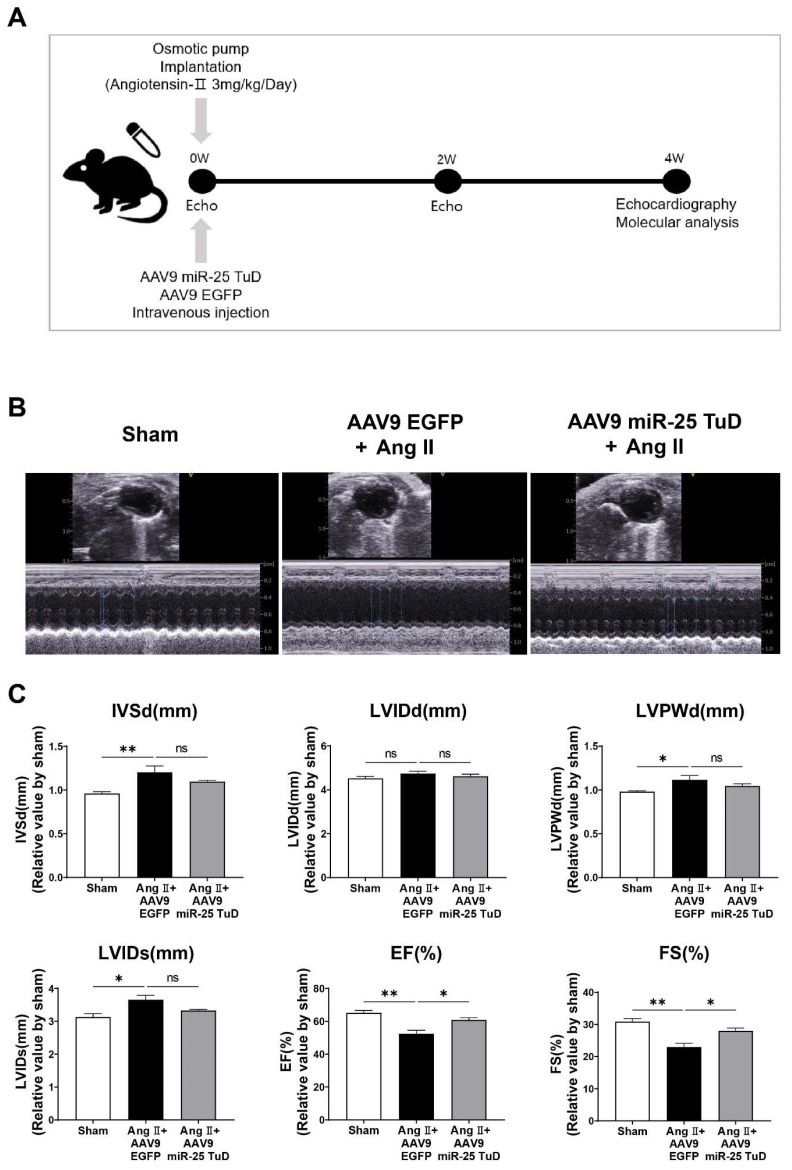
miR-25 TuD transfer prevents Ang II-induced cardiac dysfunction. (**A**) Experimental scheme for the pre-treatment of miR-25 TuD in the Ang II-infusion model. (**B**) Representative left ventricular M-mode echocardiographic images from sham, AAV9 EGFP, and AAV9 miR-25 TuD- transferred groups were obtained at 4 weeks post-injection. (**C**) Left ventricular functional parameters are shown. IVSd: interventricular septal thickness at diastole (mm), LVIDd: left ventricular internal dimension at diastole (mm), LVPWd: left ventricular posterior wall thickness at diastole (mm), LVIDs: left ventricular internal dimension at systole (mm), EF: ejection fraction (%), FS: fractional shortening (%). Data are presented as mean ± standard error of the mean (SEM). (*n* = 4 (sham-operated), *n* = 3 (AAV9-EGFP + Ang II), and *n* = 4 (AAV9-TuD + Ang II); ns: not significant, * *p* < 0.05, ** *p* < 0.01.)

**Figure 3 ijms-24-12434-f003:**
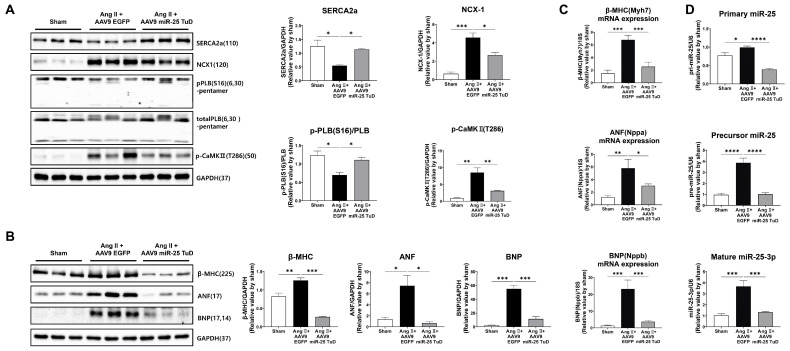
Inhibition of miR-25 normalizes calcium signaling and cardiac stress markers. (**A**) Calcium handing proteins, including SERCA2a, NCX1, p-PLB (S16), and p-CaMK II (T286), were measured via Western blot analysis and quantified using the internal control, GAPDH. Cardiac stress makers, including β-MHC, ANF, and BNP, were also measured and quantified using Western blot analysis (**B**) as well as qRT-PCR (**C**). (**D**) The expression levels of three different forms of endogenous miR-25 (primary, precursor, mature) were evaluated to confirm the long-term effect of AAV9 miR-25 TuD (qRT-PCR, *n* = 4 for each group). Data are represented as mean ± standard error of the mean (SEM). (* *p* < 0.05, ** *p* < 0.01, *** *p* < 0.001, **** *p* < 0.0001, *n* = 4 for each group.)

**Figure 4 ijms-24-12434-f004:**
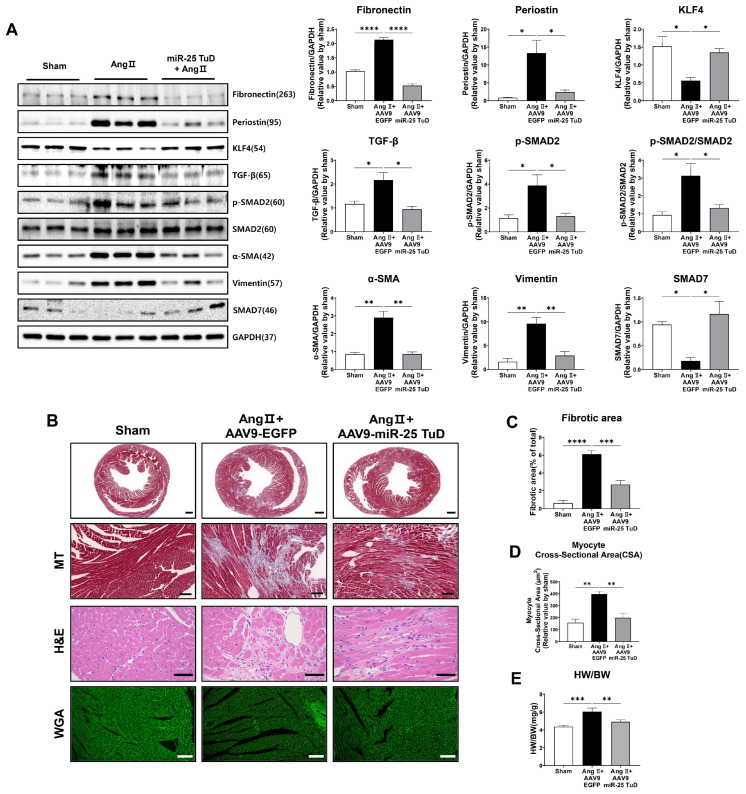
Restoration of KLF4 expression via miR-25 TuD delivery ameliorates cardiac fibrosis. (**A**) Western blotting analysis of representative cardiac fibrosis markers including Fibronectin, Periostin, KLF4, TGF-β, p-SMAD2, SMAD2, α-SMA, Vimentin, and SMAD7. (**B**) Histological assessment was conducted using Masson’s trichrome (MT), hematoxylin–eosin (H&E), and wheat germ agglutinin (WGA) staining on the sectioned heart specimens from each group. A scale bar indicates 500 μm, 100 μm, 40 μm, and 50 μm from the top to bottom, respectively. (**C**) Quantification results of the fibrotic area in each group are shown (*n* = 4 for each group). (**D**) Average cardiomyocyte cross-sectional area (at least 6 images for each group) and (**E**) heart weight (HW)-to-body weight (BW) ratios were measured (*n* = 3 (sham), 4 (AAV9-EGFP), 4 (AAV9-miR-25 TuD)). Data are presented as mean ± standard error of the mean (SEM). (* *p* < 0.05, ** *p* < 0.01, *** *p* < 0.001, **** *p* < 0.0001.)

**Figure 5 ijms-24-12434-f005:**
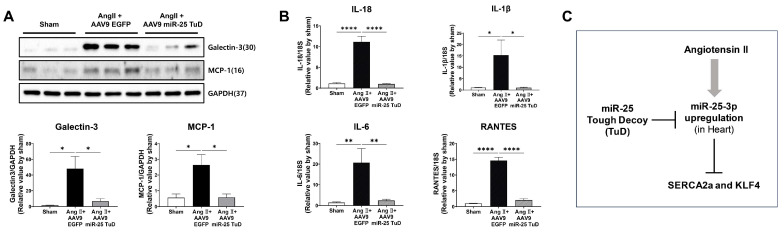
miR-25 TuD transfer prevents cardiac inflammation via KLF4 expression. (**A**) Representative inflammation markers such as Galetcin-3 and MCP-1 were quantified using Western blotting analysis. (**B**) The expression of representative inflammatory cytokines was also evaluated using qRT-PCR analysis (*n* = 4 for each group). (**C**) Schematic representation of the proposed signaling pathway of miR-25. Data are presented as mean ± standard error of the mean (SEM). (* *p* < 0.05, ** *p* < 0.01, **** *p* < 0.0001.)

## Data Availability

Not applicable.

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
