# Peer review of "Inhibition of miR-25 Ameliorates Cardiac Dysfunction and Fibrosis by Restoring Krüppel-like Factor 4 Expression"

_ijms, 2023, doi:10.3390/ijms241512434_

Round 1
Reviewer 1 Report
The present manuscript showed the role of miR-25 in cardiac dysfunction and hypertrophy. It showed that inhibiting miR-25 restores cardiac functions and reversed hypertrophy by upregulating KLF-4 expression. It also showed the correlation between miR-25 and Calcium upregulation in myocardium and showed that inhibition of miR-25 by AAV9 miR-25 TuD inhibitor will upregulate the expression of SERCA2a which is a calcium handling protein. The present study will add to the therapeutic management of hypertensive heart disease. The study is well-designed and performed. The results are supported by suitable discussion and references.
The reviewer found some minor suggestions
1. Line 51: Remove hyphen from how-ever.
2. Line 65: Remove epithelial before EMT.
3. Line 70: Remove hyphen from cardi-ac.
4. Line 83: insert o (degree) in superscript after 37.
5. Line 85, 86: supercript 4 in 104 and 2 in cm2
6. Line 83 and 90: what was the actual dose of streptomycin?
7. Line 88: Put hyphen between HEK and 293T cells.
Section: 2.5: mention the required information for qPCR as per MIQE guidelines.
8. Line 183: Remove hyphen from con-firming.
Minor improvement is required in English
Author Response
Thank you for your thoughtful comments.
Here are the detailed responses.
- Line 51: Remove the hyphen from how-ever.
Response 1: We removed the hyphen accordingly.
- Line 65: Remove epithelial before EMT.
Response 2: We removed "epithelial" from the sentence.
- Line 70: Remove the hyphen from cardi-ac.
Response 3: We removed the hyphen from it.
- Line 83: insert o (degree) in superscript after 37.
Response 4: We inserted o (degree) after 37 accordingly.
- Line 85, 86: supercript 4 in 104 and 2 in cm2
Response 5: Thank you for your comment. We corrected it accordingly.
- Line 83 and 90: what was the actual dose of streptomycin?
Response 6: Thank you for your comment. We used 1% of streptomycin and penicillin for in vitro cell culture. The material and method section was appropriately updated.
- Line 88: Put hyphen between HEK and 293T cells.
Response 7: We correctly put the hyphen between HEK and 293T cells.
Section: 2.5: mention the required information for qPCR as per MIQE guidelines.
Response Section: 2.5: Thank you for your comment. We provided a detailed protocol for qPCR method based on the MIQE guideline in the revised manuscript.
- Line 183: Remove hyphen from con-firming.
Response 8: Thank you for your comment. We removed the hyphen from it.

Reviewer 2 Report
The paper by Choulong Lee, Sunghye Cho and Dongtak Jeong (manuscript ijms-2520554) describes an in-depth in vitro and in vivo study aiming to prove inhibition by miR-25 overexpression of transcription factor Krüppel-like factor 4 (Klf4) and some of its biological consequences. Experiments have been performed in HEK293 cells by co-transfection of a plasmid containing a construct of the human Klf4 gene and firefly luciferase with plasmids containing preliminary miR-25 (pre-miR-25) and/or an antisense oligonucleotide targeting miR-25 (miR-25 TuD) and showed inhibition of Klf4 expression by pre-miR-25, while miR-25 TuD counteracted this effect, thus confirming an in silico predicted direct interaction of miR-25 with the Klf4 mRNA 3’ untranslated region (3’-UTR). Subsequent experiments on a mouse embryonic cardiomyocyte precursor cell line (H9c2) confirmed the antihypertrophic effects of the anti-miR-25 oligonucleotide in a setting of angiotensin-II-induced hypertrophic response. The angiotensin-II exposure was also performed on an animal model via subcutaneous injection from an automated implantable minipump for 4 weeks, with echocardiography assessment of left ventricular contractile performance parameters followed by histopathology and gene and protein expression assessment of hypertrophy, fibrosis and inflammation. The study is sound, original and of potential interest for a wide audience, therefore I would recommend publication in International Journal of Molecular Sciences upon minor revision according to the following comments.
1. The cardiomyocyte hypertrophy response, particularly that induced by angiotensin-II and related methods such as transverse aortic constriction, is a complex biological phenomenon involving multiple signaling pathways and molecular effectors besides specific microRNA families (miR 21, 133, 150, 195, 214). Some (already) classical studies on this topic showed the involvement of both store-operated (TRPC1,2,4,5, Orai1 in interaction with Stim1) and receptor-operated (TRPC3,6,7) calcium entry via multiple TRP channels in angiotensin-induced cardiomyocyte hypertrophy, together with recruitment of signaling pathways including the canonical TGF-beta pathway (TGF-beta -> Smad2/3, 4 - regulated by Klf members such as Klf5, 15), non-canonical TGF-beta pathways (e.g. Rho A, ROCK, SRF/MRTF), the calcineurin-NFAT pathway activated by both capacitative and receptor-operated calcium entry, all leading to activation of expression of genes leading to hypertrophy and myofibroblast transformation. Some of these mechanisms deserve being briefly commented upon within the Discussion section, and some comprehensive references may be cited, for example Eder P, Molkentin JD. TRPC channels as effectors of cardiac hypertrophy. Circ Res. 2011;108(2):265–72; Lighthouse JK, Small EM. Transcriptional control of cardiac fibroblast plasticity. J Mol Cell Cardiol. 2016;91:52–60.
2. The authors could briefly expose the variety of biological effects of the Klf family of transcription activators, and speculate if some of these effects may limit therapeutic usefulness of miR-25 targeting with antisense oligonucleotide constructs. For example, Klf4 transactivates iNOS gene expression in connection with p65 (RelA), p21Cip1/Waf1 expression in connection with p53, suppresses p53 and ornithine decarboxylase expression by competing with specificity protein 1 (Sp-1), and interacts with p300/CBP transcription coactivators. In vascular smooth muscle cells (VSMC) Klf4 acts as an anti-shear stress response gene, inhibiting thrombosis, restenosis and atherosclerosis, but also inhibits VSMC growth and proliferation and produces vasorelaxation by iNOS activation. The main role of Klf4 seems to be suppression of cell division in the presence of DNA damage, and regulating centrosome and chromosome number. It is an important regulator of the Wnt signaling pathway, controlling cell differentiation and proliferation in many tissues (also remember it is one of the four pluripotency genes used for iPSC reprogramming by Shinya Yamanaka). Generally Klf4 is an antitumorigenic factor, but in some cancers it may act as a tumor promoter (oral squamous cell carcinoma, primary breast ductal carcinoma), and it also favors epithelial hyperplasia and dysplasia, leading to squamous cell carcinoma in the skin or esophageal mucosa. Klf4 controlls macrophage polarization, upregulates ApoE (antiatherosclerotic effects), suppresses angiogenesis by regulating Notch1 activity (but in the CNS Klf4 overexpression may lead to vascular dysplasia), may promote inflammation via the NF-kB pathway in macrophages, and downregulates TNF-alpha induced VCAM-1 expression. Therefore, assessment of a putative Klf4 overexpression therapy for myocardial hypertrophy and fibrosis by miR-25 targeting requires extensive experimental work before validation.
3. Furthermore, the autors could briefly comment on the possible side effects of a miR-25 targeting therapy with antisense oligonucleotides, such as exposed for example in Crooke ST, Liang X-H, Baker BF, Crooke RM. Antisense technology: A review. J. Biol. Chem. 2021; 296:100416: interactions with proteins, interactions with RNAs, systemic toxicities (effects on complement system, clotting cascade, platelets, immunotoxicity, overactivation of the innate immune system, multiple other toxicities at cell or organ level. It is true that for the animal model used in this study the anti-miR-25 oligonucleotide was delivered with an AAV vector and expression was intracellular, but for human clinical therapy this is not a current option.
4. Some minor suggested corrections:
- page 2 lines 91-93: instead of “multiple studies had showed that KLF4 inhibits cardiac hypertrophy and fibrosis by regulating TGF-β” should be “multiple studies have shown that KLF4 inhibits cardiac hypertrophy and fibrosis by regulating TGF-β”
- Methods section, e.g. in 4.1 Cell Culture and Treatment: use superscript and subscript consistently; use °C instead of C.
No specific comments.
Author Response
I appreciate all your invaluable comments.
- The cardiomyocyte hypertrophy response, particularly that induced by angiotensin-II and related methods such as transverse aortic constriction, is a complex biological phenomenon involving multiple signaling pathways and molecular effectors besides specific microRNA families (miR 21, 133, 150, 195, 214). Some (already) classical studies on this topic showed the involvement of both store-operated (TRPC1,2,4,5, Orai1 in interaction with Stim1) and receptor-operated (TRPC3,6,7) calcium entry via multiple TRP channels in angiotensin-induced cardiomyocyte hypertrophy, together with recruitment of signaling pathways including the canonical TGF-beta pathway (TGF-beta -> Smad2/3, 4 - regulated by Klf members such as Klf5, 15), non-canonical TGF-beta pathways (e.g. Rho A, ROCK, SRF/MRTF), the calcineurin-NFAT pathway activated by both capacitative and receptor-operated calcium entry, all leading to activation of expression of genes leading to hypertrophy and myofibroblast transformation. Some of these mechanisms deserve being briefly commented upon within the Discussion section, and some comprehensive references may be cited, for example Eder P, Molkentin JD. TRPC channels as effectors of cardiac hypertrophy. Circ Res. 2011;108(2):265–72; Lighthouse JK, Small EM. Transcriptional control of cardiac fibroblast plasticity. J Mol Cell Cardiol. 2016;91:52–60.
Response 1: Thank you for your invaluable comment.
I agree with your suggestion, thus several key miRNAs involved in AngII-induced cardiac hypertrophy and remodeling are referred to in the Discussion section of the revised manuscript as follows.
Previously, several signaling molecules and miRNAs were characterized in AngII-mediated cardiac hypertrophy and remodeling in the context of calcium homeostasis. For instance, miR-1, 133a, and miR-150 are known to be negative regulators of cardiac hypertrophy through the modulation of the calcineurin-NFAT signaling pathway and p300 [35-37]. In contrast, miR-155, miR-195, and miR-214 are reported as pro-hypertrophic miRNAs by the regulation of SOCS1, MFN2, and SIRT3, respectively [38-40]. Furthermore, both store-operated and receptor-operated calcium entries via multiple TRP channels are also involved in Ang II-induced cardiomyocyte hypertrophy [41].
Cardiac fibrosis is a condition characterized by an abnormal accumulation of fibrous connective tissue in the heart. Initially, it results from a repair process in response to injury or stress, such as chronic inflammation or damage due to conditions like hypertension, heart attacks, or cardiomyopathy. However, excessive and uncontrolled deposition of fibrous tissue in the heart can lead to a stiffening and thickening of the cardiac muscle, disrupting its normal function [42]. Multiple miRNAs and signaling molecules were also identified to regulate cardiac fibrosis. For example, miR-21 has been validated as an activator of cardiac fibroblasts post-MI, subsequently eliciting cardiac fibrosis [43, 44]. In addition, Wang et al. reported that miR-92a [45] and miR-195 [46] act as transcriptional regulators of SMAD7, an inhibitor of α-SMA, which is a well-established marker of myofibroblast activation. miR-34a also modulates cardiac fibrosis after MI via targeting SMAD4 [47]. Furthermore, it is also known that the canonical and non-canonical TGF-b pathways have been intensively studied to understand the underlying molecular mechanism of cardiac remodeling and fibrosis [48]. However, the role of miR-25 has not been clearly elucidated in terms of cardiac hypertrophy and fibrosis. For this reason, we sharply focused on the role of miR-25 in the present study.
- The authors could briefly expose the variety of biological effects of the Klf family of transcription activators, and speculate if some of these effects may limit the therapeutic usefulness of miR-25 targeting with antisense oligonucleotide constructs. For example, Klf4 transactivates iNOS gene expression in connection with p65 (RelA), p21Cip1/Waf1 expression in connection with p53, suppresses p53 and ornithine decarboxylase expression by competing with specificity protein 1 (Sp-1), and interacts with p300/CBP transcription coactivators. In vascular smooth muscle cells (VSMC) Klf4 acts as an anti-shear stress response gene, inhibiting thrombosis, restenosis and atherosclerosis, but also inhibits VSMC growth and proliferation and produces vasorelaxation by iNOS activation. The main role of Klf4 seems to be suppression of cell division in the presence of DNA damage, and regulating centrosome and chromosome number. It is an important regulator of the Wnt signaling pathway, controlling cell differentiation and proliferation in many tissues (also remember it is one of the four pluripotency genes used for iPSC reprogramming by Shinya Yamanaka). Generally Klf4 is an antitumorigenic factor, but in some cancers it may act as a tumor promoter (oral squamous cell carcinoma, primary breast ductal carcinoma), and it also favors epithelial hyperplasia and dysplasia, leading to squamous cell carcinoma in the skin or esophageal mucosa. Klf4 controlls macrophage polarization, upregulates ApoE (antiatherosclerotic effects), suppresses angiogenesis by regulating Notch1 activity (but in the CNS Klf4 overexpression may lead to vascular dysplasia), may promote inflammation via the NF-kB pathway in macrophages, and downregulates TNF-alpha induced VCAM-1 expression. Therefore, assessment of a putative Klf4 overexpression therapy for myocardial hypertrophy and fibrosis by miR-25 targeting requires extensive experimental work before validation.
Response 2: Once again, we appreciate all your great insight and comments on the role of Klf4.
As recommended, we briefly described the various biological effects of Klf4 in the Discussion section.
Klf4 act as a tumor suppressor as well as a promoter since it contains both transcriptional activation and suppression domains. Therefore, the function of KLF4 in most types of tumors is controversial and confusing (cell Death Discovery volume 9, Article number: 118 (2023)). This applies to the cardiovascular system as well. For example, as you mentioned, Klf4 can transactivate iNOS gene expression in vascular smooth muscle cells (VSMC), which produces a vasodilation effect. In addition, it also inhibits VSMC growth and proliferation in some cases. Therefore, the restoration of Klf4 expression by the inhibition of miR-25 should be carefully examined before it applies, which is the main obstacle to the use of miR-25 targeting therapy. However, to date, it is generally accepted that Klf4 restoration generally mitigates pathological cardiac hypertrophy and prevents detrimental cardiac dysfunction.
- Furthermore, the authors could briefly comment on the possible side effects of a miR-25 targeting therapy with antisense oligonucleotides, such as exposed for example in Crooke ST, Liang X-H, Baker BF, Crooke RM. Antisense technology: A review. J. Biol. Chem. 2021; 296:100416: interactions with proteins, interactions with RNAs, systemic toxicities (effects on complement system, clotting cascade, platelets, immunotoxicity, overactivation of the innate immune system, multiple other toxicities at cell or organ level. It is true that for the animal model used in this study the anti-miR-25 oligonucleotide was delivered with an AAV vector and expression was intracellular, but for human clinical therapy this is not a current option.
Response 3: Thank you for your invaluable comment.
To date, we have not observed the side effect of antisense oligonucleotide therapy against miR-25 yet in our setting, but we added a paragraph for the possible limitation of the current strategy and technical difficulties using antisense oligonucleotide and AAV vector system in the discussion section as follows.
Lastly, there are several limitations to performing translational research with the current approach used in this manuscript. First, although AAV has been reported to show minimal immune responses, virtually neutralizing antibodies (NAbs) that recognize all AAV serotypes can be found in a large proportion of the human population (ranging from 30 to 60%). Therefore, AAVs have been genetically engineered to avoid this issue by increasing AAV transduction efficiency or vector tropism (Nature Reviews Genetics volume 21, 255–272 (2020)). Second, there is a possible off-target expression of anti-oligonucleotide delivery even with cardiotropic AAV serotype 9. In order to overcome this issue, we may use a tissue-specific promoter or a newly reconstructed cardiotropic vector. Nevertheless, AAV-mediated gene delivery was recently approved for the treatment of several inherited diseases, and it is the most promising delivery method in pre-clinical and clinical settings.
- Some minor suggested corrections:
- page 2 lines 91-93: instead of “multiple studies had showed that KLF4 inhibits cardiac hypertrophy and fibrosis by regulating TGF-β” should be “multiple studies have shown that KLF4 inhibits cardiac hypertrophy and fibrosis by regulating TGF-β”
Response 4-1: Thank you for your comment. We correctly updated the sentence as recommended.
- Methods section, e.g. in 4.1 Cell Culture and Treatment: use superscript and subscript consistently; use °C instead of C.
Response 4-2: Thank you for your comment. We corrected all typos.
